# A Game-Theoretic Rent-Seeking Framework for Improving Multipath TCP Performance †

**Shiva Raj Pokhrel** [1,*] **and Carey Williamson** [2]

1   School of Information Technology, Deakin University, Geelong, VIC 3220, Australia
2   Department of Computer Science, University of Calgary, Calgary, AB T2N 1N4, Canada
*   Correspondence: shiva.pokhrel@deakin.edu.au
†   This is an extended version of our work presented at IFIP Performance 2020, Milan, Italy, 2–6 November 2020. As per the conference policy, an eight-page extended abstract appears in the ACM SIGMETRICS Performance Evaluation Review.

**Abstract:** There is no well-defined utility function for existing multipath TCP algorithms. Therefore, network utility maximization (NUM) for MPTCP is a complex undertaking. To resolve this, we develop a novel condition under which Kelly's NUM mechanism may be used to explicitly compute the equilibrium. We accomplish this by defining a new utility function for MPTCP by employing Tullock's rent-seeking paradigm from game theory. We investigate the convergence of no-regret learning in the underlying network games with continuous actions. Based on our understanding of the design space, we propose an original MPTCP algorithm that generalizes existing algorithms and strikes a good balance among the important properties. We implemented this algorithm in the Linux kernel, and we evaluated its performance experimentally.

**Keywords:** multipath TCP; congestion control; Tullock rent seeking; normalized equilibrium

## 1. Introduction

The ongoing development of 5G and gigabit WiFi solutions will considerably improve throughput and delay performance on future wireless networks. These developments offer new possibilities for enhancing existing applications by using multipath TCP (MPTCP) [1] to exploit multiple available network interfaces. Multipath TCP seamlessly aggregates the capacities of multiple access technologies, and it is now an integral part of ATSSS [2] (access traffic steering, switching, and splitting) for 5G [3].

A challenging design issue that arises is that of best using WiFi and non-WiFi 5G seamlessly [4] across a variety of applications and platforms. This is a challenge, since different applications impose different requirements and constraints for successful MPTCP usage.

One example application is video streaming. This seems like an ideal use case for MPTCP, since it has been difficult to scale existing services to meet the growing traffic demands of mobile users. For some existing applications, such as online video conferencing, high-resolution video still suffers from jitter, unwanted glitches, and frozen screens when used over mobile networks.

Augmented reality (AR) is another emerging application that faces similar challenges due to bandwidth demands. AR allows users with nomadic wireless devices to seamlessly access information by overlaying graphics and digital information upon their perception of the physical world. This enables immersive AR/VR applications, but requires dependable, low-latency, and high-bandwidth data communication.

Vehicular networking is yet another application that could benefit from MPTCP [5]. For example, autonomous vehicles may need to send 3D views of their environment to remote monitoring servers under extremely tight deadlines to ensure safe operation. Similarly, unmanned aerial vehicles, such as delivery drones, have stringent communication requirements for navigating their dynamically changing environment in a cooperative

fashion. One example of this would be using live video from multiple drones to anticipate obstacles ahead and avoid risky blind spots while on the move.

These aforementioned applications have three key similarities. First, they are throughput-intensive. Second, they require low end-to-end delay. Third, they need reliable data packet transport.

Although the mobile/wireless networking standards are evolving to support higher capacity requirements, existing transport-layer protocols (e.g., TCP, UDP, QUIC, and MPTCP) rarely provide adequate support for low-latency and high-throughput applications, especially in wireless networks. One oft-cited reason for this is the tendency of TCP congestion control to induce excessive buffering delay at the network bottleneck. Further compounding this problem are the vagaries of wireless channels, which often exhibit non-deterministic behavior in signal quality.

Aside from the application aspects (5G/WiFi) mentioned above, the main motivation of this research is to advance the theoretical knowledge in the overhaul of MPTCP, which requires a major departure from the existing design. These theoretical challenges are discussed in the following.

### Motivation and Contributions

Kelly et al. [6,7] designed a distributed congestion control framework for obtaining a network equilibrium with an objective of achieving *proportional fairness*. This model holds as long as the TCP sources react to the sum of the congestion prices in their paths. Moreover, when the TCP sources react to the maximum of the prices in the network paths, the system instead achieves *max–min fairness*.

Our main insight is that Kelly's approach of "*transforming the fairness problem into a game between TCP users*" shares several features with the Tullock *rent-seeking* game [8]. The main differences will be discussed later.

In coupled MPTCP algorithms, it is known that, if one path fails to meet the loss and delay constraints, then the subflows on other paths suffer [9–11]. Therefore, the competition between MPTCP subflows can be considered as a game with common coupled constraints. In such games, when a player fails to achieve the constraints common to other players, all other players will also be penalized. Such games can, therefore, be viewed as cooperative in the objectives of meeting common constraints and non-cooperative in the context of utilities. We exploit the *rent-seeking* game framework for studying the contests between subflows, thus providing insights into the design of *cooperative MPTCP* algorithms.

Masson et al. [12] used a similar framework to model contests over time for maximizing visibility in social networks. We observe that the *rent-seeking* game has the potential to model MPTCP transmissions over multiple network paths. Each MPTCP subflow may bid its proposed packet-sending rate $\lambda_i$ before transmitting. Then, each MPTCP subflow will pay a price $p_i$ based on its bid. Based on this price, they eventually obtain the throughput $\theta_i$ proportional to their bid.

In a realistic network setting, since the achievable bandwidth across the network paths is finite, this approach will lead to (infinitely) many equilibria. Unfortunately, the difficulty in designing a mechanism for selecting a desired equilibrium point and, thus, maximizing the utility is a serious drawback with this approach. One possible solution to this is to use the *Karush–Kuhn–Tucker* (KKT) conditions to solve a similar (relaxed) version of the game, rather than the exact constrained game. This relaxed game has the same equilibria as the original game, but is easier to solve.

Using Kelly's theory, the Lagrange multipliers can be interpreted as *pseudo-prices* associated with the network path, and the equilibrium will satisfy the (desired) coupled constraints. However, the KKT conditions alone are not sufficient for our needs, since these do not guarantee that the *pseudo-price* per unit bandwidth is the same for all contending subflows. Rather, this approach increases the already large state space of our MPTCP system model and makes it unscalable. More specifically, it becomes intractable, since once *pseudo-*

*prices* (Lagrange multipliers) are obtained, they depend not only on the policy adopted by an MPTCP subflow, but also on the corresponding strategies of other coupled subflows.

Our idea is to find a *pseudo-price* that is globally constant per unit resource across all network paths. Recall that such an equilibrium with a global fixed *pseudo-price* is the desired *normalized* equilibrium for our MPTCP dynamics over the multipath network. Therefore, the Tullock game framework is used to redesign MPTCP and analyze the existence and uniqueness of the equilibrium $(\lambda^\star, p^\star)$. Moreover, we prove that, starting from any initial point $(\lambda(0), p(0))$, the trajectory $(\lambda(t), p(t))$ generated by the cooperative MPTCP algorithm over multiple paths converges to a unique solution with a globally stable equilibrium $(\lambda^\star, p^\star)$ [13].

To summarize, the main contributions of our work are the following:

- We design a utility function for MPTCP based on Tullock's rent-seeking game.
- We prove that this utility function implies a unique and stable solution to the MPTCP optimization problem.
- We examine the convergence of no-regret learning in the underlying network games.
- We provide extensive experimental results showing the performance of our MPTCP algorithm.

The rest of this paper is organized as follows. Section 2 provides an overview of our MPTCP modeling approach and situates it in the context of prior work. Section 3 discusses the details of our approach, including the game theory framework and the equilibrium properties of our system. Section 4 extends our model to consider multiple MPTCP flows and no-regret learning. Section 5 presents the experimental results of the evaluation of the performance of our MPTCP solution using network emulation experiments. Section 7 concludes the paper.

## 2. Model Overview

This section provides the background context for our work by summarizing prior work on network optimization and MPTCP design. The section concludes with a conceptual overview of our MPTCP modeling approach using Tullock's rent-seeking framework (RSF).

### 2.1. Background and Related Work

Of particular relevance to this work are the core foundational works on charging and rate control for elastic traffic and their analyses reported in [6,7,13–21]. Kelly's approach in [7] presented price-taking users to attain system optimality or efficiency. However, when the users are price-anticipating (or strategic), the users exhibit rent-seeking behavior. This problem then falls within Tullock's rent-seeking framework [13], and there is an efficiency loss [17,18]. For a finite-capacity single-path TCP user's case where the path is shared by all TCP users/flows, there are a few important works, viz. Hajek and Gopalakrishnan [16], Johari and Tsitsiklis [17], and Altman et al. [13,14], among others.

The analyses in [16,17] deal with the so-called '*posted prices*' setting in which all users perceive the same price on the path. The works [13,14] deal with differential pricing. For a discussion on the advantage of differential pricing over posted prices, see the implementation–theoretical perspective explained in [19].To this end, the existence and uniqueness of the Nash equilibrium were proved independently in [16,17] and by using Rosen [18] in [13] or variations of Rosen [14]. The authors of [17] also dealt with network settings where agents can use multiple paths.

Kelly's theory on *network utility maximization* (NUM) [6] provides a rigorous foundation for the design of efficient TCP algorithms. It has proven useful for understanding the structural properties of networks under end-to-end congestion control and for the design of optimal TCP algorithms. For path selection in multipath congestion control, Key et al. [22] developed another novel approach in order to study the potential benefits of coordinated congestion control. This enables users to select (or reselect) routes by adopting the notion of a Nash equilibrium.

There are two distinct steps in most existing optimization frameworks for TCP congestion control design. First, one develops a mathematical model for the resource allocation problem (e.g., fluid and/or queueing model [23]). Second, one designs an algorithm to solve the model (e.g., convex optimization based on its properties) [23,24].

A limitation of such NUM-based approaches is that they often assume a stable and steady-state network. As a result, they may lead to TCP algorithms that struggle in the presence of highly fluctuating and time-varying characteristics, such as those in wireless networks (e.g., 5G and next-generation WiFi). Specifically, the rapid variation in the capacity of these wireless channels may cause their state to change before the TCP algorithm converges on its optimal solution.

In NUM and current TCP design, utility is typically a function of the instantaneous packet-sending rate (i.e., throughput). This approach leads to bandwidth-sharing networks [6], which are suitable for bulk data transfer flows over slowly varying channels. In addition, all of the TCP algorithms proposed (and deployed) have strictly concave utility functions, implying a unique stable equilibrium.

However, the case for MPTCP is much more complex. This is because the existence of an underlying utility function depends on the design of the coupling function between the subflows (see Table 1 in [23]). For many well-known MPTCP algorithms, such as LIA [4], OLIA [9], and BALIA [23], a utility function does not even exist. Therefore, *utility maximization for MPTCP is highly challenging*.

Existing MPTCP congestion control algorithms [4,9,10,23] have used common coupled constraints (e.g., loss/delay) to represent the channel characteristics and congestion conditions of the network paths (see Table 1 for a concise summary of the TCP window update operations for LIA, OLIA, and BALIA; as mentioned earlier, if one path fails to meet the constraints, the subflows on other paths may also suffer [9–11]).

**Table 1.** Summary of MPTCP congestion control for LIA, OLIA, and BALIA.

| MPTCP Algorithms | Increase on Path $i$ | Decrease on Path $i$ |
|---|---|---|
| **LIA** [4] $\alpha = \dfrac{\max\left(W_i/\tau_i^2\right)}{\left(\sum_i \frac{W_i}{\tau_i}\right)^2}\sum_i W_i$ | For each acknowledgment, $\Delta(W_i) = \min\left(\frac{1}{W_i}, \frac{\alpha}{\sum_i W_i}\right)$ | For each packet loss, $W_i \leftarrow \frac{W_i}{2}$ as regular TCP |
| **OLIA** [9] $\alpha_i = \begin{cases} \frac{1/|\mathcal{I}|}{|\mathcal{C}|}, & i \in \mathcal{C} \\ \frac{1/|\mathcal{I}|}{|\mathcal{X}|} & i \in \mathcal{X}, |C| > 0 \\ 0 & \text{otherwise} \end{cases}$ | For each acknowledgment, $\Delta(W_i) = \left(\frac{\alpha_i}{W_i} + \frac{W_i/\tau_i^2}{\left(\sum_i \frac{W_i}{\tau_i}\right)^2}\right)$ | For each packet loss, $W_i \leftarrow \frac{W_i}{2}$ as regular TCP |
| **BALIA** [23] $\alpha_i = \frac{\max\{\lambda_k\}}{\lambda_i},\ \lambda_i = \frac{W_i}{\tau_i}$ | For each acknowledgment, $\lambda_k = \left(\sum_{k \in \mathcal{I}} \frac{W_k}{\tau_k}\right)^2$ $\Delta(W_i) = \frac{\lambda_i}{\tau_i(\sum \lambda_k^2)}\frac{1+\alpha_i}{2}\frac{4+\alpha_i}{5}$ | For each packet loss, $W_i \leftarrow W_i - \frac{W_i}{2}\min\left(\alpha_i, \frac{3}{2}\right)$ |

Moreover, the throughput achieved by a subflow depends in subtle ways upon the throughputs of all other subflows belonging to the same MPTCP connection [4,10,15]. When packets are sent via diverse wireless paths, they can arrive at the receiver out of order or may be lost due to channel impairments. Lost packets must be retransmitted and out-of-order packets must be buffered for resequencing so that they can be delivered to the application in proper order. Both of these artifacts affect subflow throughput because of buffering delays and the need for retransmissions at either the data-link layer or the MPTCP level [15].

At a conceptual level, almost all coupled MPTCP congestion control designs focus on minimizing reordering delay and head-of-line (HOL) blocking. They provide both load balancing and congestion control by coupling the delays and losses across all subflows. Based on our understanding of the design space, we propose a new MPTCP algorithm

that generalizes existing algorithms and strikes a good balance among these properties. We implemented this algorithm in the Linux kernel, and we evaluated its performance experimentally by using our prototype.

*2.2. Our Proposed Scheme*

We now summarize our proposed MPTCP algorithm. The key idea is to schedule the packets via multiple paths at different rates in such a way that they balance the delay across all paths and are, therefore, more likely to arrive at the receiver in order. More specifically, an MPTCP congestion control mechanism with heterogeneous network paths will perceive different utilities on each path.

In essence, we strive to harmonize the performance achieved across multiple paths by opportunistically exploiting path dynamics to provide seamless MPTCP performance. The primary advantage of our method is that it implicitly addresses all known issues that limit MPTCP performance (i.e., HOL blocking, reordering latency, route heterogeneity). Furthermore, it enables the pooling of multipath resources in a way that emulates the performance of a single harmonized path.

One way to achieve utility maximization is to formulate a Tullock rent-seeking game [8,25] and use this to equalize the delays across all paths in order to ameliorate reordering delay. A plausible approach is to set equal delay for all subflows (across all paths) and determine the corresponding sending rates that jointly satisfy the delay constraints and load-balancing objectives.

DELAY-BALANCED MPTCP (RSF-DB): In RSF-DB, each source has a set of paths and a predefined target delay $d > 0$, which is fixed and equal for all paths. The source maintains a congestion window $W_i$ for each path $i$ and measures its forward delay $q_i$ and roundtrip time $\tau_i$ for each path. Here, $q_i$ is the queueing plus service delay from the source to the sink on path $i$, where $K$ is the number of paths. Then, the window adaptation in the congestion avoidance phase is:

○   For each packet loss on path $i$, update $W_i \leftarrow W_i/2$;
○   For each new acknowledgment received on path $i$, increase $W_i$ using $W_i \leftarrow W_i + \Delta(W_i)$, where:

$$\Delta(W_i) = \frac{1}{W_i} \min\left\{1, \frac{d - q_i}{q_i}\right\}. \tag{1}$$

This algorithm is designed to adjust the window size dynamically so that $q_i$ converges toward the target delay $d$ in the steady state. Note that $q_i$ is not necessarily strictly bounded by $d$, so the "window increase" could, in some cases, be negative if the situation warrants.

A concise summary of the window update operations for three well-known MPTCP algorithms (LIA, OLIA, and BALIA) is shown in Table 1.

Similarly, we designed a complementary solution called LOSS-BALANCED MPTCP (RSF-LB), in which each MPTCP source has a set of paths and a predefined target loss rate. The key idea here is to balance the loss *rate* (i.e., lost packets per second) on each path to ameliorate the potential effects of heterogeneity in the packet loss *probabilities*, $l_i$, on different paths. For RSF-LB, we replace (1) with

$$\Delta(W_i) = \frac{1}{W_i} \min\left\{1, \frac{\mu - \lambda_i l_i}{\lambda_i l_i}\right\} \tag{2}$$

so that the instantaneous $\lambda_i l_i$ converges toward the target loss rate $\mu$ packets/sec in the steady state.

We extend Kelly's framework [6] by exploiting the well-known Tullock rent-seeking game [8] to design a cooperative MPTCP with common coupled constraints. We consider a non-linear dynamical system model for an MPTCP connection over $K$ network paths, as seen in Equation (3):

$$\dot{y} = f(y), \quad y(0) = y_0, \tag{3}$$

where $f(\mathbf{y}) = \mathbf{0}$ has a unique solution. In this equation,

$$\mathbf{y} = (\boldsymbol{\lambda}, \mathbf{p}); \ \boldsymbol{\lambda} = (\lambda_1, \lambda_2, \ldots \lambda_K)$$

the $\lambda_i$ values denote the packet-sending rates, and

$$\mathbf{p} = (p_1, \ldots p_K)$$

denote the congestion prices (based on the delay $q_i$ or loss $l_i$) for the respective paths. Then, motivated by the generalized MPTCP analysis in [23], we specify the following fluid model:

$$\dot{\lambda}_i = k_i(\boldsymbol{\lambda})(\phi_i(\boldsymbol{\lambda}) - \mathbf{p})_{\lambda_i}^+; \ \dot{p}_i = \gamma_i(\lambda_i - \theta_i)_{p_i}^+ \tag{4}$$

for analyzing MPTCP (Equations (3) and (4) in [23]). Here, $\gamma_i > 0$ is a positive gain, $k_i(\boldsymbol{\lambda})$ is a vector of positive gains determining the dynamic properties of the system, and $\phi_i(\boldsymbol{\lambda})$ determines the equilibrium properties.

Our objective is to find a *pseudo-price* that is globally fixed per unit of resource across all network paths. The equilibrium in Equation (4), along with this special *pseudo-price*, is known as a *normalized* equilibrium for MPTCP dynamics over multiple networks (Section 2 in [13]). Therefore, a successful idea developed recently by Altman et al. [13] can be used to analyze the existence and uniqueness of the equilibrium $(\boldsymbol{\lambda}^\star, \mathbf{p}^\star)$. Moreover, starting from any initial point $(\boldsymbol{\lambda}(0), \mathbf{p}(0))$, the trajectory $(\boldsymbol{\lambda}(t), \mathbf{p}(t))$ generated by the cooperative MPTCP algorithm converges to a unique and globally stable equilibrium $(\boldsymbol{\lambda}^\star, \mathbf{p}^\star)$ (see Lemma 2 later in the paper).

In summary, the approach in our model is as follows. We use an existing fluid modeling approach for MPTCP to quantify its interactions over the network paths as a dynamic system [23]. We then implement the Lagrangian method to solve the dynamic system. However, for the Lagrangian approach to succeed, we require the existence of a well-defined utility function with certain properties to guarantee the stability. Therefore, we establish a utility function that satisfies these properties by adopting the idea from Tullock's rent-seeking game [8,13], which we describe in the next section.

## 3. Modeling Framework

The details of the background of the Tullock rent-seeking game and its equilibrium properties were explained with the proposed MPTCP in our preliminary work [1]. In this section, we discuss the essential remaining details for completeness.

### 3.1. Rent-Seeking Game Framework

Rent-seeking is a concept from game theory in economics. Unlike profit-seeking, which tries to increase total wealth, rent-seeking tries to increase one's own share of existing wealth without increasing the total wealth. Specifically, it is an attempt to obtain a reward R (rent) by manipulating the environment in which the activities occur, rather than by creating new wealth. One example is lobbying and/or bribery of government officials, in which the efforts (e.g., time and/or money) spent may result in favorable outcomes (e.g., tax laws, corporate rebates), albeit with potentially reduced economic efficiency. Other examples include taxi-licensing fees, bridge tolls, and bidding in loss–load curves [26], all of which result in reallocation of resources.

Strategies for *rent-seeking* games were studied by Gordon Tullock in 1967 and form a good basis for studying MPTCP flows. We use this rent-seeking approach to design our utility function below.

We consider an MPTCP connection in a network setting with *K* paths and one subflow on each path. Let $\lambda_i = W_i/\tau_i$ (strictly positive $\lambda_i > 0$) represent the sending rate for the *i*th subflow ($1 \leq i \leq K$), and let $p_i$ represent the corresponding price for that subflow. Then, assuming that the sending rates on each path are the actions to be chosen and that the

connection throughput is concave, the individual throughput (payoff) obtained from path $i$ can be represented as [13]:

$$\theta_i = V\left(\lambda_i \Big/ \sum_{j=1}^{K} \lambda_j\right),\tag{5}$$

where $V(.)$ is a concave function. Observe that the throughput $\theta_i$ comes at the cost of $\hat{p}\lambda_i$ (where $\hat{p}$ is a constant). With this formulation, the utility for the MPTCP subflow over path $i$ is structurally similar to that of a Tullock rent-seeking game [13], for which:

$$U_i(\lambda) = \lambda_i \Big/ \sum_{j=1}^{K} \lambda_j - \hat{p}\lambda_i.\tag{6}$$

**Lemma 1** (Existence and Uniqueness). *The utility $U_i(\lambda)$ of an MPTCP subflow $i$ is concave in its action $\lambda_i$ and is continuous in the corresponding actions of other subflows. Further, for $\hat{p} > 0$ (strictly positive pricing), the trajectory $(\lambda(t), p(t))$ generated by the cooperative MPTCP algorithm has a unique Nash equilibrium in pure policies.*

The proof of the concavity and continuity of $U_i(\lambda)$ follows from direct calculation. The existence and uniqueness can be established along the lines of Rosen [18] and Altman [13], respectively.

In contrast to Kelly's approach [6], the subflows of an MPTCP connection can have an action on one path (such as an increase or decrease in $\lambda_i$) that is not dependent on the actions over other paths. Further, similarly to [6], the capacity of the system is limited in real communication networks. Therefore, the aggregate network bandwidth can be bounded by a finite constant, say $B$. As a result,

$$\sum_{j=1}^{K} \lambda_j \leq B.\tag{7}$$

However, with this additional capacity constraint, *the uniqueness of the Nash equilibrium* given by Lemma 1 collapses, and there can be (infinitely) many equilibria [27].

*3.2. Equilibrium Properties*

We now discuss the application of our new utility function from Equation (6) to the solution of our dynamic system defined in Equation (4).

Let $(\lambda^\star, p^\star)$ be the equilibrium of the aforementioned dynamical system and let $\lambda^\star_{(-i)}$ be the action vectors of all of the MPTCP subflows, except for the one along path $i$. Now, by using KKT, since $U_i(\lambda)$ is concave for each $\lambda_i$, there exists a Lagrange multiplier $L_i(\lambda^\star_{(-i)})$ such that $\lambda^\star_i$ maximizes the Lagrangian:

$$Q_i(\lambda_i) = U_i(\lambda, \mathbf{x}_{(-i)}) - L_i(\mathbf{x}_{(-i)})\left(B - \sum_{j=1}^{K} \lambda_j\right);\tag{8}$$

$$\text{and}\qquad L_i(\mathbf{x}_{(-i)})\left(B - \sum_{j=1}^{K} \lambda_j\right) = 0.\tag{9}$$

Equation (9) is obtained by using the complementary property. In addition, observe in Equation (8) that the Lagrangian $Q_i(.)$ replaces the utility $U_i(.)$.

The Lagrange multipliers are interpreted as *pseudo-prices* of the paths. If a price is set on path $i$ while other players are at equilibrium, then the subflow on path $i$ pays $\lambda_i L_i(\lambda^\star_{(-i)})$ for its use of the bandwidth. In this case, $\lambda^\star$ is an equilibrium of our non-linear dynamical system with the capacity constraints. However, the pricing is not tractable, since it may

vary from one subflow to another (for similar use of the network resources). In addition, it depends on the equilibrium selected.

Therefore, our main objective is to find a *constant* Lagrange multiplier $L$ that is independent of $i$. Furthermore, $L$ should be independent of the actions taken by any of the MPTCP subflows, along with an associated equilibrium $\lambda^\star$ for our new relaxed game. If we can find such a constant Lagrange multiplier $L$, the trajectory $(\lambda(t), p(t))$ generated by our cooperative MPTCP algorithm converges to a unique and globally stable *normalized* equilibrium $(\lambda^\star, p^\star)$. Moreover, our interest is to establish the existence and uniqueness of the *normalized* equilibrium. Therefore, we investigate the *diagonally strict concave utility* functions of our dynamical system (given by Equation (1)), since it is known that these lead to a unique global equilibrium (Theorem 4 in [18]).

### 3.3. Diagonal Strict Concavity

Consider a game with $K$ subflows of an MPTCP connection. The strategy space is $S \subset \mathbb{R}$, where $S$ is a bounded set, and subflow $i$'s utility function is $U_i : S^K \to \mathbb{R}$. Rosen's condition [18] for uniqueness of the Nash equilibrium in a $K$-player game states that the equilibrium is unique when: (i) $U_i(\lambda)$ is concave in subflow $i$'s own action; (ii) There exists a vector $z$ of non-negative numbers such that the function $\rho(\lambda, z) := \sum_{i=1}^{K} z_i U_i(\lambda)$ is diagonally strictly concave.

To define the concept of *diagonal strict concavity*, we first compute the pseudo-gradient of $\rho(.)$:

$$g(\lambda, z) = \begin{bmatrix} z_1 \frac{\partial U_1(\lambda_1, \lambda_{-1})}{\partial \lambda_1} \\ z_2 \frac{\partial U_2(\lambda_2, \lambda_{-2})}{\partial \lambda_1} \\ \vdots \\ z_K \frac{\partial U_K(\lambda_K, \lambda_{-K})}{\partial \lambda_1} \end{bmatrix} \tag{10}$$

where $\lambda_{-i} = \sum_{j=1(\neq i)}^{K} \lambda_j$. Then, the function $\rho(.)$ is said to be *diagonally strictly dominant* in $z \in S$ for fixed $z \leq 0$ if for every $\lambda_0, \lambda_1 \in S$

$$(\lambda_0 - \lambda_1)' g(\lambda_1, z) + (\lambda_1 - \lambda_0)' g(\lambda_0, z) > 0. \tag{11}$$

CONDITION $C_0$: A sufficient condition for the function $\rho(.)$ to be diagonally strictly concave is that the matrix $[G'(\lambda, z) + G(\lambda, z)]$ is negative definite for $z \in S$: here $G(\lambda, z)$ is the Jacobian of the pseudo-gradient $g(\lambda, z)$ with respect to $\lambda$.

CONDITION $C_1$: A sufficient condition for the existence of a unique global equilibrium is that $\rho(.)$ is diagonally strictly concave for some $z$ (i.e., $C_0$ holds).

Note that CONDITION $C_1$ is essential for the stability of RSF MPTCP. Observe that $C_1$ is obtained directly from Theorem 4 in Rosen (Theorem 4 in [18]). We acknowledge that both prior works [13,18] nicely established the existence, uniqueness, and stability of the underlying dynamical system for the Tullock *rent-seeking* game. We implement these ideas in our work to design a cooperative framework for MPTCP. Furthermore, we consider a practical operational model, in which the existence of the utility function, and its interaction with multipath network constraints is one of the building blocks.

**Lemma 2** (Stability). *Assume that $C_0$ and $C_1$ both hold, the utility function $U(.)$ exists, and the existence and uniqueness of the equilibrium $(\lambda^\star, p^\star)$ is guaranteed. Then, starting from any initial point $(\lambda(0), p(0))$, the trajectory $(\lambda(t), p(t))$ generated by the cooperative MPTCP algorithm converges to a unique solution, and Equation (1) has a globally stable solution given by $(\lambda^\star, p^\star)$.*

First, we provide a proof that a Tullock *rent-seeking* game with a finite capacity constraint has a diagonally strictly concave utility function and, thus, a unique *normalized* equilibrium.

**Proof.** We have

$$
g(\lambda, z) = \begin{bmatrix} z_1 \dfrac{\sum_{i=2}^K \lambda_i}{\left(\sum_{i=1}^K \lambda_i\right)^2} \\[2ex] z_2 \dfrac{\sum_{i=1(\neq 2)}^K \lambda_i}{\left(\sum_{i=1}^K \lambda_i\right)^2} \\[1ex] \vdots \\[1ex] z_K \dfrac{\sum_{i=1(\neq K)}^K \lambda_i}{\left(\sum_{i=1}^K \lambda_i\right)^2} \end{bmatrix} ; \text{ and}
$$

$$
G_{i,j} = \frac{\partial}{\partial \lambda_j}\left(\frac{\partial}{\partial \lambda_i} \frac{z_i \lambda_i}{\left(\sum_{i=1}^K \lambda_i\right)^2}\right)
$$

For the matrix $[G'(\lambda, z) + G(\lambda, z)]$,

$$
G_{i,j} + G_{j,i} = \begin{cases} -4 z_i \lambda_{-i} \big/ \left(\sum_{i=1}^K \lambda_i\right)^3, & \text{if } i = j \\[2ex] \dfrac{z_i(\lambda_i - \lambda_{-i}) + z_j(\lambda_j - \lambda_{-j})}{\left(\sum_{i=1}^K \lambda_i\right)^3} & \text{otherwise.} \end{cases} \tag{12}
$$

Next, we take a column vector $A = \begin{bmatrix} a_1 \dots a_K \end{bmatrix}^{\mathrm{T}}$, such that $[G'(\lambda, z) + G(\lambda, z)]$ is negative definite if $A'[G'(\lambda, z) + G(\lambda, z)]A < 0, \forall A, A \neq 0, i.e.,$

$$
\begin{aligned}
& A'[G'(\lambda, z) + G(\lambda, z)]A \\
&= \sum_{i=1}^K \left(\left(\sum_{j=1(\neq i)}^K a_i a_j \frac{z_i(\lambda_i - \lambda_{-i}) + z_j(\lambda_j - \lambda_{-j})}{\left(\sum_{i=1}^K \lambda_i\right)^3}\right)\right. \\
& \left. - a_i^2 4 z_i \lambda_{-i} \big/ \left(\sum_{i=1}^K \lambda_i\right)^3\right)
\end{aligned} \tag{13}
$$

Therefore, for all $z_i = 1$,

$$
A'[G'(\lambda, z) + G(\lambda, z)]A = -F \Big/ \left(\sum_{i=1}^K \lambda_i\right)^3 \tag{14}
$$

where $F$ is given by

$$
\begin{aligned}
&= \sum_{i=1}^K \left(4 a_i^2 \lambda_{-i} + \left[\sum_{j=1(\neq i)}^K a_i a_j \big((\lambda_{-i} - \lambda_i) \right.\right. \\
& \left.\left. + (\lambda_{-j} - \lambda_j)\big)\right]\right) \\
&= \sum_{i=1}^K \left(4 a_i^2 \left(\left(\sum_{i=1}^K \lambda_i\right)^3 - \lambda_i\right)\right. \\
& \left. + \left[\sum_{j<i}^K 4 a_i a_j \left(\left(\sum_{i=1}^K \lambda_i\right)^3 - \lambda_i - \lambda_j\right)\right]\right) \\
&= 4 \sum_{i=1}^K \left(a_i^2 \sum_{j=1(\neq i)}^K \lambda_i + \left[\sum_{j>i)}^K a_i a_j \sum_{k=1(\neq i, \neq j)}^K \lambda_j\right]\right) \\
&= 4 \sum_{i=1}^K \lambda_i \left[\sum_{j=1(\neq i)}^K a_j^2 + a_j \sum_{k>j(\neq i)}^K a_k\right]
\end{aligned} \tag{15}
$$

Observe in Equation (15) that F is positive for $\lambda > 0$; therefore, the matrix $[G'(\lambda, z) + G(\lambda, z))]$ is negative definite. As a result, $U(\lambda)$ is diagonally strictly concave. See [13] for more background. □

In the following, we provide further details on stability and our interpretation of utility maximization. Given the normalized equilibrium and the concave utility, the remainder of the proof of Lemma 2 is provided in Section 3.3.3.

### 3.3.1. Lyapunov Stability Theory

We apply Lyapunov's theorem to prove the stability of the nonlinear dynamic system represented by Equation (3). We make use of the following Lyapunov stability theorem to test the stability of the system.

LYAPUNOV'S THEOREM FOR STABILITY: If there exists a Lyapunov function $L(\lambda)$ of a system, then $\lambda = 0$ is a stable equilibrium point in the sense of Lyapunov: (i) If $dL(\lambda)/dt < 0, 0 < ||\lambda|| < r$ for some $r$, i.e., if $dL(\lambda)/dt$ is locally negative definite, then $\lambda = 0$ is an asymptotically stable equilibrium point; (ii) if $L(\lambda)$ is positive definite on the entire state space and has the property that $|L(\lambda)| \to \infty$ as $||\lambda|| \to \infty$, and the derivative $dL(\lambda)/dt$ is negative definite on the entire state space (except at the equilibrium point), then the equilibrium point is globally asymptotically stable.

A nonlinear system is asymptotically stable around its equilibrium if it satisfies the aforementioned Lyapunov theorem. The first condition of the theorem requires that the state trajectory can be confined to an arbitrarily small 'ball' of radius $r$ centered at the equilibrium. This is called stability in the sense of Lyaponov.

### 3.3.2. Utility Maximization Interpretation

The well-known approach for an MPTCP-based dynamic system is to associate a utility function $U_i(\lambda_i)$ with each user $i$ and interpret the sources and network to be executing distributed algorithms to maximize the aggregate utility of the users. The primary idea is to connect the equilibrium points of the dynamical systems with the solution of some optimization problem. Therefore, the nonlinear dynamical system can be viewed as the one that solves the utility maximization problem in a distributed manner. For example, $(\lambda_i^*, p_i^*)$ is an equilibrium of the dynamical system of MPTCP users if and only if $\lambda_i^*$ is optimal for

$$\max \sum_n^N \sum_{i \in \mathcal{K}} U_i(\lambda_i) \; s.t. \; y_i \leq B_i \tag{16}$$

where $p_i^*$ is the Lagrange multiplier or pseudo-price associated with the link at the optimal $\lambda_i^*$. Here, $y_i \leq B_i$ means that the aggregate traffic ($y_i$) at each link is less than the capacity $B_i$ of the link.

In fact, all of the existing TCP algorithms proposed in the literature have concave utility functions implying a unique stable equilibrium [23]. Furthermore, we conclude that, for our MPTCP, given $p_i$, we have a unique $\lambda_i$ given by (see Equation (6))

$$\lambda = U_i'^{-1}(p_i). \tag{17}$$

### 3.3.3. Stability of the Dynamical System

We make use of a Lyapunov function for Lasalle's invariance principle to prove the global asymptotic stability of the dynamical system. Let $\partial p := p - p^\star$ and $\partial \lambda := \lambda - \lambda^\star$. The function

$$L(\lambda, p) = \sum_i \left( \int_{\lambda_i^\star}^{\lambda_i} \left( \frac{z - \lambda_i^\star}{\bar{k}_i(z)} \right) dz + \gamma_i \partial p_i^2 \right) \tag{18}$$

is a Lyapunov function. To prove Lemma 2, we extend the ideas from [23], and therefore, we include the essential steps only.

**Proof.** Differentiating $L(\boldsymbol{x}, \boldsymbol{p})$ with respect to $t$,

$$\dot{L}(\boldsymbol{\lambda}, \boldsymbol{p}) = \sum_i \left( \frac{\partial \lambda_i \dot{\lambda}_i}{\bar{k}_i(\lambda_i)} + \gamma_i \partial p_i \dot{p}_i \right).$$

If $\partial \boldsymbol{\lambda} \neq 0$, then, using (4),

$$\frac{\partial \lambda_i \dot{\lambda}_i}{\bar{k}_i(\lambda_i)} = \partial \lambda_i (\phi_i(\boldsymbol{\lambda}) - p_i)^+_{\lambda_i} \leq \partial \lambda_i (\phi_i(\boldsymbol{\lambda}) - p_i)$$

$$= \partial \lambda_i (\phi_i(\boldsymbol{\lambda}) - \phi_i(\boldsymbol{\lambda}^\star) - \partial p_i). \tag{19}$$

Furthermore,

$$\sum_i \left( \partial \lambda_i \dot{\lambda}_i / \bar{k}_i(\lambda_i) \right)$$

$$\leq \partial \boldsymbol{\lambda}^T (\boldsymbol{\phi}(\boldsymbol{\lambda}) - \boldsymbol{\phi}(\boldsymbol{\lambda}^\star)) - \partial \boldsymbol{\lambda}^T \partial \boldsymbol{p} < -\partial \boldsymbol{\lambda}^T \partial \boldsymbol{p}. \tag{20}$$

Similarly,

$$\sum_i \gamma_i \partial p_i \dot{p}_i = \sum_i \partial p_i (\lambda_i - \theta_i)^+_{p_i} \leq \sum_i \partial p_i (\lambda_i - \theta_i)$$

$$\leq \sum_i \partial p_i \partial \lambda_i = \partial \boldsymbol{p}^T \partial \boldsymbol{\lambda}. \tag{21}$$

Now,

$$\text{if } \partial \boldsymbol{\lambda} \neq 0, \text{ then } \dot{L}(\boldsymbol{\lambda}, \boldsymbol{p}) < \partial \boldsymbol{p}^T \partial \boldsymbol{\lambda} - \partial \boldsymbol{\lambda}^T \partial \boldsymbol{p} = 0;$$

$$\text{if } \partial \boldsymbol{\lambda} = 0, \text{ then } \dot{L}(\boldsymbol{\lambda}, \boldsymbol{p}) = 0. \tag{22}$$

Therefore, $\dot{L}(\boldsymbol{\lambda}, \boldsymbol{p}) \leq 0$ and $L(\boldsymbol{\lambda}, \boldsymbol{p})$ *is indeed a Lyapunov function.*

Let $\Gamma := (\boldsymbol{\lambda}(t), \boldsymbol{p}(t))$ be a set of trajectories on which $\dot{L}(\boldsymbol{\lambda}(t), \boldsymbol{p}(t)) = 0$ for all $t \geq 0$. The only trajectory in $\Gamma$ is the trivial trajectory $(\boldsymbol{\lambda}, \boldsymbol{p}) \equiv (\boldsymbol{\lambda}^\star, \boldsymbol{p}^\star)$. Therefore, Lasalle's invariance principle implies that $(\boldsymbol{\lambda}^\star, \boldsymbol{p}^\star)$ is globally asymptotically stable.

Observe that $\dot{L} \equiv 0$ implies $\partial \boldsymbol{\lambda} \equiv 0$, i.e., any trajectory in $\Gamma$ must have $\boldsymbol{\lambda}(t) = \boldsymbol{\lambda}^\star$ for all $t \geq 0$. Hence, $(\boldsymbol{\lambda}, \boldsymbol{p}) \equiv (\boldsymbol{\lambda}^\star, \boldsymbol{p}^\star)$ is the only trajectory in $\Gamma$. □

## 4. Multiple MPTCP Connections

Now, we extend our solution to the case of $N$ MPTCP sources sharing $K$ paths. Each subflow $i$ of an MPTCP connection $n$ over the $K$ paths has a sending rate $\lambda_i^n$, and the following constraint should hold:

$$\sum_{n=1}^{N} \sum_{i=1}^{K} \lambda_i^n \leq B. \tag{23}$$

The throughput obtained by the $n$th MPTCP connection is the sum of the throughputs for all of its subflows and is given by:

$$\theta^n = \sum_{i=1}^{K} \theta_i^n(\boldsymbol{\lambda^n}). \tag{24}$$

where $\boldsymbol{\lambda^n}$ is the vector $[\lambda_1^n, \lambda_2^n, \ldots, \lambda_K^n]$ and

$$\theta_i^n(\boldsymbol{\lambda^n}) = V\left( \frac{\lambda_i^n}{\sum_{m=1}^{N} \sum_{j=1}^{K} \lambda_j^m} \right) \text{(recall Equation (5))}. \tag{25}$$

Therefore, the utility of the $n$th MPTCP connection is:

$$U^n(\boldsymbol{\lambda}) = \sum_{i=1}^{K} \left( \frac{\lambda_i^n}{\sum_{m=1}^{N} \sum_{j=1}^{K} \lambda_j^m} - \hat{p} \lambda_i^n \right). \tag{26}$$

Observe that the net utility in Equation (26) is computed as the sum of the utilities of each subflow. This decoupling approximation is motivated by the findings in realistic network settings [10], as each network path $i$ has its own capacity (bandwidth) and prices (independent of each other); therefore, let $C$ be the vector with the $i$th element being a finite constant $C_i$. Then, $B = \sum C_i$, and for each $i$, we have the following constraint:

$$\sum_{n=1}^{N} \lambda_i^n \leq C_i. \tag{27}$$

Due to this constraint, the target loss and delay of RSF MPTCP are attainable iff the *aggregate achievable loss/delay is lower than the target*.

Observe that when applying the KKT conditions to the optimal response at the normalized equilibrium, there are $K$ Lagrange multipliers. Clearly, our objective is to find a vector of $K$ Lagrange multipliers such that the Nash equilibrium for the relaxed game will be an equilibrium for the primary game, where the constraints will be satisfied along with the conditions for the complementary equation. If diagonal strict concavity holds for each subflow and/or for each path separately, then Theorem 4 in [18] guarantees the existence and uniqueness of the equilibrium.

*Learning for Continuous Actions*

For learning continuous actions of MPTCP, we propose the adoption of the generally practiced dual-averaging method of Nesterov [28] in our new game-theoretic setting. Intuitively, the main idea of the learning process is as follows. At each stage of the process, the window adaptation process of path $i$ generates a new estimate of the individual update of the throughput function at the current action profile, which is possibly subject to loss, channel impairments, and delay uncertainties. With this estimate, one may move into the dual space (for gradients) and translate the performance back into the primal space to select the next action, as shown in Figure 1.

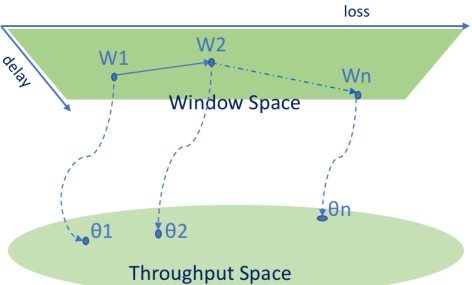

**Figure 1.** Primal dual mapping: illustrative representation of dual averaging.

Starting with some arbitrary estimate, our approach can be captured using recurrence relations as follows (using Equation (5)):

$$\theta_{i,n,m} = V(\lambda_{i,n,m}) \approx V(W_{i,n,m}), \tag{28}$$
$$W_{i+1,n,m} = W_{i,n,m} + \gamma_m \hat{\theta}_{i,n,m+1} \tag{29}$$

where $m$ represents the step of the process, $\hat{\theta}_{i,n,m+1}$ is the estimate of the individual throughput gradient of the $i$th subflow of the $n$th MPTCP. To this end, the core components of dual averaging for learning continuous actions are the gradient estimates and the mapping, which determines the actions of the MPTCPs as players. For the MPTCPs' gradient obser-

vations, we assume that each MPTCP subflow $i$ has an underlying feedback mechanism that responds with a prediction of the throughput gradients at its current profile. However, such information can be imperfect for several reasons: (i) Predictions are susceptible to errors; (ii) predictions may suffer from channel impairments; (iii) throughput functions may be the stochastic expectations of some random variable ($\theta_i(W_i) = \mathbb{E}[\hat{\theta}(W_i; p_i)]$ [29]), which the MPTCP source can only observe via the realized sample path of gradients. Observe that Equations (17) and (18) hold for both RSF-LB and RSF-DB, which were proposed earlier (see Equations (1) and (2)).

Cognizant of such issues, we follow along the lines of [30] and focus on the erroneous feedback mechanism of the form:

$$\hat{\theta}_{i,n,m+1} = \theta_i(W_{n,m+1}) + \xi_{i,n,m+1}. \tag{30}$$

Observe that the noise process $\xi_m = (\xi_{i,n,m})_{i \in \mathcal{N}}$ is an $L^2$-bounded martingale difference (with respect to filtration) from the history $(\mathcal{F}_m)_{m=1}^{\infty}$ of $W_{n,m}$ such that $\xi_{m+1}$ is not $\mathcal{F}_m$ measurable (however, $\xi_m$ is $\mathcal{F}_m$ measurable). Therefore, $\xi_m$ satisfies the following *zero-mean* and *finite mean-square error* properties:

$$\mathbb{E}[\xi_{m+1}|\mathcal{F}_m] = 0; \quad \forall m \in \{1, 2, \dots\}. \tag{31}$$

$$\mathbb{E}[||\xi_{m+1}||^2|\mathcal{F}_m] \leq \sigma^2; \quad \forall m \in \{1, 2, \dots\} \text{ and } \sigma \geq 0. \tag{32}$$

With Equations (31) and (32), the gradient estimates of each path are unbiased and bounded in the mean square conditionally. Specifically,

$$\begin{aligned} \mathbb{E}[\hat{\theta}_{m+1}|\mathcal{F}_m] &= \theta(W_m), \\ \mathbb{E}[||\hat{\theta}_{m+1}||^2|\mathcal{F}_m] &\leq \Theta_*^2; \quad \forall m \in \{1, 2, \dots\} \text{ and } \Theta_* > 0. \end{aligned} \tag{33}$$

Such an interpretation is generic in the incorporation of a broad range of erroneous processes in our dynamical system of MPTCP data flows, including all compactly supported log-normal, Gaussian, and exponential distributions. Furthermore, both of the above theories can also be relaxed for finite-order moments considering a small bias.

Based on the foregoing discussion, an appropriate choice for $V_n$ is $y_n \leftarrow arg\ max_{\theta_i}(y_i, \theta_i)$. However, such an approach that outputs actions that are closely aligned with $y_n$ has two main problems:

- Due to uncertainty in the channel dynamics, the *arg max* action choice can be too aggressive.
- Convergence of the dual-averaging (DA) approach is challenging, since the output will always be an extreme point.

Therefore, we suggest to soften the approach by using a regularized mapping:

$$y_n \leftarrow arg\ max_{\theta_i}\{(y_i, \theta_i) - h_i(\theta_i)\}, \tag{34}$$

where the regularizing function $h : \mathcal{C} \to \mathbb{R}$ is a penalty function on a compact convex subset $\mathcal{C}$ of a finite-dimensional norm if: (i) $h$ is strictly convex; (ii) the limit of $h(\theta)$ as $\theta$ approaches $B$ equals $h(B)$ (i.e., $h$ is continuous). Observe that these conditions are also comparable to the MPTCP conditions reported in [23]. Furthermore, the mapping induced by $h$ can be interpreted as [30]:

$$V(y) = arg\ max_{\theta_i}\{(y_i, \theta_i) - h_i(\theta_i)\}. \tag{35}$$

In what follows, we assume that each MPTCP subflow has its own individual $h_i$ value and focus on the interplay between dual and primal parameters (i.e., the outputs $\theta_i$ and the actions $W_i$, respectively). More precisely, from an MPTCP source perspective, it can be viewed as an aggregate penalty function $h(\theta) = \sum_i h_i(\theta_i)$. As shown in Algorithm 1, the learning process can be observed as a multi-agent version of gradient descent with (lazy) projections. The mapping '$V_n \circ \theta_i$' is most often defined as smooth and the regularized

best response in a game-theoretical context; see [30,31] for detailed discussions (McKelvey et al. [27] referred to $V_n$ as a 'quantal response function' in finite-player games).

---

**Algorithm 1** Abstract View of the Dual-Averaging Algorithm

---

 1: **procedure** INITIALIZE
 2: Step size ($\gamma_m$), initial windows ($W_i$)
 3: **end procedure**
 4: **procedure** UPDATE
 5:     **while** $n \leq N$ **do**
 6:         **for** each MPTCP subflow $i$ **do**
 7:             Choose action: use Equation (17)
 8:             Estimate gradient: use Equation (19)
 9:             Take gradient step: use Equation (18)
10:         **end for**
11:     **end while**
12: **end procedure**

---

One main advantage of dual averaging in concave games is that this leads to *no regret* as long as the steps of Algorithm 1 are correctly picked [32]. While certain, within DA, the average payoff of each subflow corresponds to the most effective reverse case (note that this ignores changes in the behavior of other subflows owing to an alteration in the action selected by one subflow). With relevant insights from these observations and our ongoing foundational works [33,34], we plan to improve on this worst-case guarantee and obtain appropriate convergence results for the actual play series triggered by DA in future work.

## 5. Evaluation

In this section, we present the performance evaluation of our new MPTCP algorithm in different network scenarios. We describe the network settings used in the experiments and then present our experimental results for each scenario.

### 5.1. Experimental Setup

We evaluated our *cooperative MPTCP* algorithm based on the rent-seeking framework (RSF) by comparing it with three other well-known MPTCP algorithms: LIA [4], OLIA [9], and BALIA [23]. The latter three algorithms are all available in an experimental Linux implementation (C. Paasch et al., "Multipath TCP in the Linux Kernel", available from https://www.multipath-tcp.org (released: 2 November 2017)) of MPTCP v0.93 by Paasch et al. We used this Linux implementation as the basis for our experiments and added implementations for our RSF MPTCP, including both delay-balanced (RSF-DB) and loss-balanced (RSF-LB) versions.

The test environment in our lab consisted of three laptops, all running Ubuntu Linux 16.04. As shown in Figure 2, one of these was set up as an HTTP server, while the other two were clients. The laptops were connected to an Ethernet switch, each via two 1 Gbps interfaces, thus creating two separate paths for our experiments. In particular, each of our MPTCP connections had two subflows in our experiments (for simplicity [23]). We used the NetEm (NetEm: https://wiki.linuxfoundation.org/networking/netem, accessed on 13 May 2022) network emulator to control the bandwidth, delay, and packet loss on the Gigabit Ethernet links in order to emulate the characteristics of wireless paths. We used $d = 100$ ms and $\mu = 4$ packets/s in all of our experiments.

The performance evaluation of the *MPTCP* algorithms was carried out with the following settings. We used five MPTCP connections (each with two subflows) between the server and the two clients. The server ran a single MPTCP implementation at a time for each experiment. The packet traffic was generated by a large file transfer from the HTTP server, where the file size ranged from 2 to 10 MB. Packet traces were captured

using *tcpdump* (https://www.tcpdump.org, accessed on 13 May 2022) and analyzed with *wireshark* (https://www.wireshark.org, accessed on 13 May 2022).

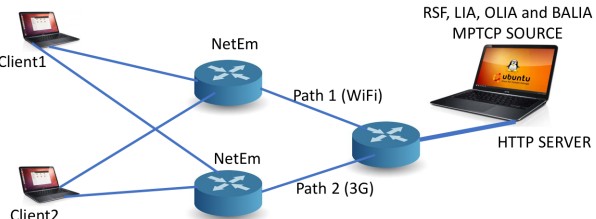

**Figure 2.** Experimental setup: Ethernet links emulate the characteristics of WiFi and 3G paths.

Table 2 provides a concise summary of our results, which are presented and explained in the following subsections. We check for the target loss ($\mu$) and delay ($d$) in all of our experiments.

**Table 2.** Summary of the throughput results from the network emulation experiments.

| ID | Emulated Network Scenario | Regular TCP | MPTCP Algorithm (Mbps) | | | |
|----|---------------------------|-------------|------|------|-------|------|
| | | | LIA | OLIA | BALIA | RSF |
| 0 | Single Path | 1.58 | 1.53 | 1.57 | 1.58 | 1.58 |
| 1 | Homogeneous Paths: No Loss | 1.58 | 2.52 | 2.58 | 2.54 | 2.53 |
| 2 | Homogeneous Paths: 2% Loss | - | 0.88 | 0.98 | 0.98 | 1.98 |
| 3 | Homogeneous Paths: 10% Loss | - | 0.31 | 0.35 | 0.39 | 0.62 |
| 4 | Heterogeneous Paths: Bandwidth | - | 1.88 | 1.88 | 1.98 | 2.89 |
| 5 | Heterogeneous Paths: Delay | - | 2.59 | 2.69 | 2.69 | 4.59 |
| 6 | Heterogeneous Paths: Loss | - | 3.13 | 3.10 | 3.12 | 4.48 |
| 7 | Short-Lived Flows | 3.42 | 5.51 | 6.21 | 7.02 | 8.67 |
| 8 | Path Failure | 2.21 | 3.88 | 3.97 | 4.19 | 5.71 |
| 9 | Competing TCP Flows | 0.25 | 0.66 | 0.67 | 0.61 | 1.20 |

*5.2. Baseline Results*

For the verification and validation of our experimental environment, we started with two baseline scenarios.

### 5.2.1. Observation 0

Scenario 0 used a single-path network, for which all TCP versions should perform the same. The first row of results in Table 2 confirms this observation, which is summarized in the rightmost column of the table.

### 5.2.2. Observation 1

Scenario 1 used two homogeneous paths with no packet loss at all. In this scenario, our RSF MPTCP should perform the same as LIA. The next row of results in Table 2 confirms this observation, as summarized in the table. In fact, we observe the exact same results for both RSF-DB and RSF-LB MPTCP.

*5.3. Homogeneous Paths*

The next set of experiments considered homogeneous paths, but with non-zero packet loss. In these scenarios, RSF MPTCP started to differentiate itself from the other MPTCP algorithms.

For Scenario 2, we set the capacity (10 Mbps), delay (150 ms), and Bernoulli packet loss probability (2%) for the links to be the same for both paths. We then studied the impacts on the RSF-DB throughput when downloading files of different sizes (2 MB $\leq$ size $\leq$ 8 MB) with different MPTCP algorithms, each operating under the same settings.

Figures 3 and 4 illustrate the throughput and fairness results, respectively. Figure 3 shows that, when the size of the file is 4 MB, our proposed *RSF MPTCP* achieves almost double the throughput of LIA, OLIA, and BALIA. Figure 4 shows that the corresponding Jain fairness index stays quite close to unity. Hence, the throughput is highly improved by using our design without any adverse impact on fairness.

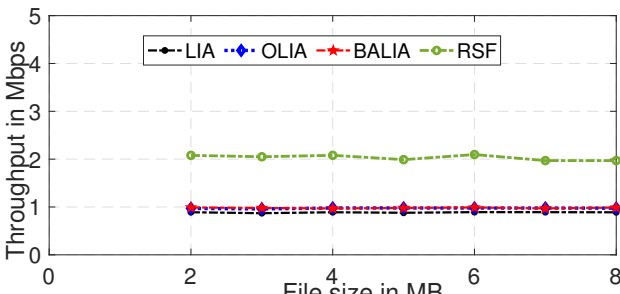

**Figure 3.** Throughput results for different file sizes in Scenario 2: homogeneous paths with 10 Mbps bandwidth, 150 ms delay, and 2% random packet loss.

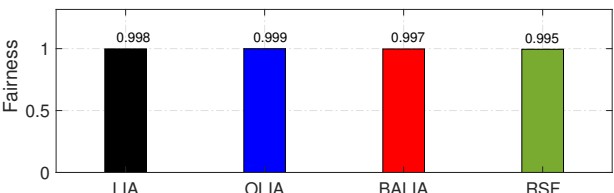

**Figure 4.** Fairness results for 4 MB files in Scenario 2: homogeneous paths with 10 Mbps bandwidth, 150 ms delay, and 2% random packet loss.

We also considered Scenario 3 with a much higher Bernoulli packet loss probability (10%) on both paths. While the throughput declined for all MPTCP algorithms, our RSF MPTCP still provided much higher throughput than the others.

Observation 2/3

The results from Scenario 2 and Scenario 3 lead to the following observation. The proposed *RSF MPTCP* outperformed all other MPTCP algorithms in terms of throughput and achieved similar fairness.

*5.4. Heterogeneous Paths*

The next set of experiments considered heterogeneous network paths.

Scenario 4 had heterogeneous paths that differed in their bandwidth. We set the delay (150 ms) and random packet loss probability (4%) to be the same for both paths. We set the bandwidth to a fixed value of 20 Mbps for one path while varying the capacity of the other one from 10 to 80 Mbps. We then measured the RSF-DB performance when downloading a 10 MB file using different MPTCP algorithms.

Figure 5 shows the impact on performance. In general, the throughputs achieved by the existing MPTCP algorithms were fairly low and remained almost constant, even with the increasing bandwidth on the second path. That is, they did not exploit the additional bandwidth well. In contrast, the RSF-DB algorithm showed consistent improvement in throughput under the same network settings. When packet losses were frequent, the existing MPTCP algorithms suffered a lot, but our algorithm was able to discover heterogeneous paths quickly and improved the load balancing across paths.

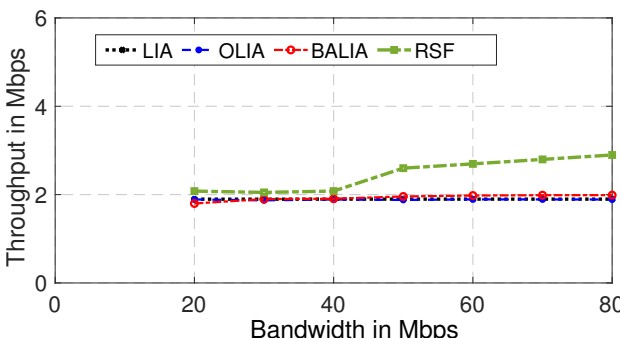

**Figure 5.** Throughput results for 10 MB file transfer in Scenario 4: heterogeneous bandwidth paths. One path was 20 Mbps, while the bandwidth of the other one was varied. Both paths had 150 ms delay and 4% random packet loss.

### 5.4.1. Observation 4

The results from Scenario 4 are summarized as follows. The proposed *RSF MPTCP* outperformed the other MPTCP algorithms when the paths had heterogeneous bandwidths.

Next, Scenario 5 considered paths with heterogeneous delay characteristics. We set equal bandwidths for the two paths (20 Mbps) and the same random packet loss probability (3%) for both paths. We fixed the delay for one of the paths at 20 ms and then varied the delay of the other path from 20 to 100 ms. As before, we observed the RSF-DB throughput when downloading a 10 MB file.

### 5.4.2. Observation 5

Figure 6 illustrates the overall impact on throughput, which can be summarized as follows. The throughput of all MPTCP algorithms (including RSF) decreased with increasing delay, as expected. However, the decline was quite gradual for RSF MPTCP and quite pronounced for the other MPTCP algorithms.

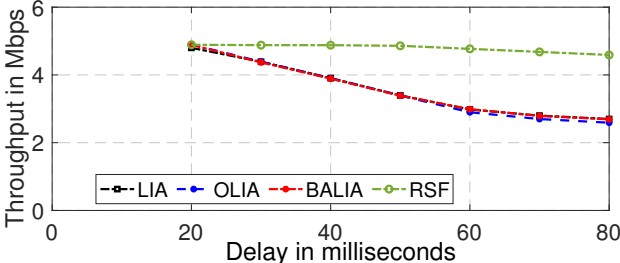

**Figure 6.** Throughput results for a 10 MB file transfer in Scenario 5: heterogeneous delay paths. One path had 20 ms delay, while the delay of the other one was varied. Both paths had 20 Mbps bandwidth and 3% random packet loss.

This result was mainly due to the delay required for re-transmission of lost packets, as well as the buffering delay when re-sequencing out-of-order packets received at the client. In contrast, the balanced pricing strategy adopted in our design made RSF MPTCP superior to the others in an environment with heterogeneous delay and non-negligible packet losses.

Scenario 6 examined heterogeneous paths with different loss characteristics. We set the packet loss probability to a fixed value of 3% for one path while varying the packet loss probability for the other path. Both paths had equal bandwidth (20 Mbps) and delay (150 ms).

Figure 7 depicts the throughput performance when downloading a file with a size of 10 MB. The results show that, with increasing packet loss, the throughput achieved by the existing MPTCP algorithms decreased. In contrast, the RSF-LB algorithm outperformed them and provided robust performance, even with increasing channel errors. This was because most existing algorithms significantly reduced congestion windows after detecting

packet losses. However, the proposed RSF MPTCP could dynamically determine the best way to adjust the congestion windows using the balanced pricing approach.

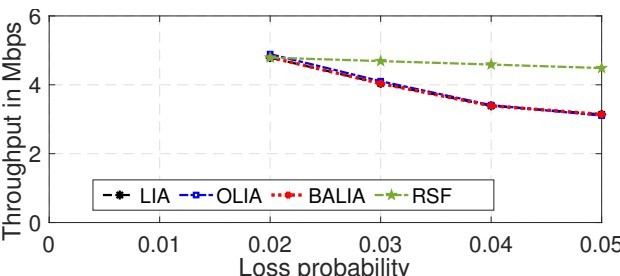

**Figure 7.** Throughput results for a 10 MB file transfer in Scenario 6: heterogeneous loss paths. One path had 3% packet loss while the loss probability of the other one was varied. Both paths had 20 Mbps bandwidth and 150 ms delay.

### 5.4.3. Observation 6

This experiment can be summarized as follows. The proposed MPTCP algorithm was robust and outperformed the others under adverse channel conditions.

More importantly, we observed that, in the case of heterogeneous paths, the target loss and delay were mostly satisfied by the proposed RSF MPTCP where possible (i.e., bounded by Equation (27)).

### 5.5. Additional Experiments

The remaining experiments considered different networking scenarios to evaluate the responsiveness and robustness of RSF MPTCP.

In Scenario 7, we used iPerf3 to study the performance of short-lived MPTCP flows. For the network setup, we used RSF-DB and set equal bandwidth (20 Mbps), delay (150 ms), and loss (2%) for both paths. For the traffic, we generated short-lived flows with randomly generated durations between 1 and 30 s. These flows arrived in the network according to a Poisson arrival process (arrival rate = 8) and departed at random times, as well.

### 5.5.1. Observation 7

Figure 8 shows the throughput perceived by small flows of different durations. The results indicate the following: The proposed RSF MPTCP was robust in a highly dynamic environment in which MPTCP flows joined and left the system frequently.

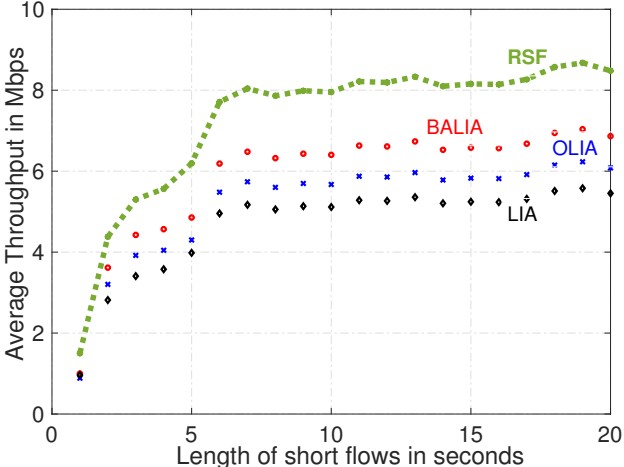

**Figure 8.** Throughput results for Scenario 7: short-lived flows. Both paths had 20 Mbps bandwidth, 150 ms delay, and 2% packet loss probability. Short-lived flows arrived and departed at random times according to Poisson processes.

The results in Figure 8 illustrate RSF MPTCP's ability to handle the frequent establishment and termination of flows. This feature reflects the increased responsiveness of our MPTCP algorithm to network changes. The two main enhancements in our design, namely, balanced pricing and adaptive congestion control, enabled this responsiveness.

In Scenario 8, we investigated a case in which one of the two paths suddenly became unavailable (e.g., WiFi path disappeared, but the LTE connection was still fine). The results are shown in Figure 9, where we compare the throughput obtained by the intermittent subflows of different MPTCPs.

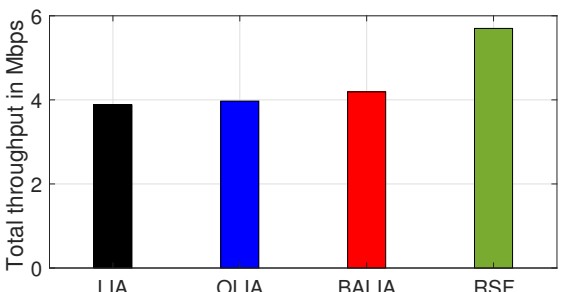

**Figure 9.** Throughput results for Scenario 8: path failure. Both paths had 20 Mbps bandwidth, 150 ms delay, and 2% packet loss probability. One path (WiFi) became unavailable at 50 s.

### 5.5.2. Observation 8

RSF MPTCP performed well in the face of a dynamically changing number of subflows, and it outperformed the other MPTCP algorithms. When the number of subflows changed over time in the realistic network scenario, we observed a sharp drop in performance at 50 s with all algorithms (including RSF MPTCP). However, our RSF-LB algorithm detected the failed path very quickly. This mechanism made the RSF MPTCP algorithm highly responsive to changes in network paths. The other existing MPTCP algorithms were all quite sluggish in detecting and abandoning the failed path.

### 5.5.3. Observation 9

Scenario 9 investigated fairness between TCP and MPTCP. Specifically, we considered a scenario of five MPTCP connections when one of the paths was shared with five regular (single-path) TCP connections.

The results obtained by using Scenario 9 are shown in Figure 10. Figure 10 illustrates the throughput perceived by both TCP and MPTCP connections, which demonstrated the following: RSF MPTCP was quite friendly towards TCP while providing better throughput than the other MPTCP algorithms.

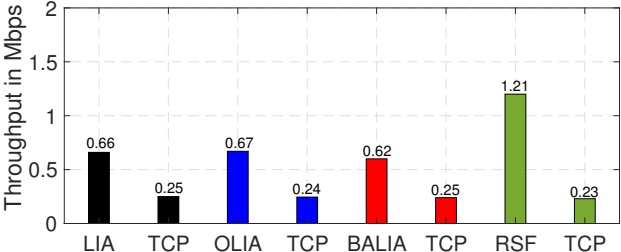

**Figure 10.** Throughput results for Scenario 9: competing TCP flows. Both paths had 20 Mbps bandwidth, 150 ms delay, and 2% packet loss probability. One path had five additional regular TCP flows, as well.

Figure 10 shows that when RSF-DB MPTCP was used, the throughput achieved by regular TCP flows was comparable to that obtained while competing with the other MPTCP algorithms. However, the corresponding MPTCP connection had much higher throughput. More specifically, when compared to the other MPTCP algorithms, the performance of TCP

when competing with cooperative MPTCP flows was slightly lower, but the throughput gain for MPTCP was very high. The throughput gain rose because of RSF MPTCP's ability to exploit the capacity of both available paths. The reason for the slightly reduced TCP throughput needs further investigation, and this remains for future work. The observation with RSF-LB was also similar.

### 5.6. WiFi-Based Experiment

We performed another experiment in a real WiFi testbed with settings similar to those of Scenario 5, but using two WiFi network interfaces. For both paths, the bandwidths were 54 Mbps and the delays were 8 ms. Packet losses were determined empirically based on the characteristics of the WiFi channel. We again used five MPTCP connections and measured the average throughput of the individual connections. The results are shown in Figure 11.

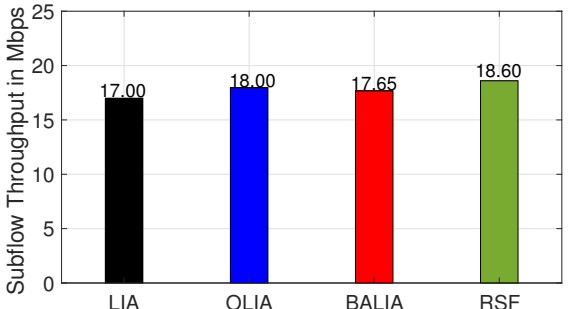

**Figure 11.** Throughput results for Scenario 9 with RSF-DB: WiFi. Both paths had 54 Mbps bandwidth and 8 ms delay. The packet loss probability was low.

In this experiment, both the delay and the packet loss probability were very low, so the throughput advantages of RSF-DB MPTCP were small (but still measurable) compared to the other MPTCP algorithms. Note that our RSF MPTCP coincided with LIA under deterministic path conditions and zero packet loss. However, these properties seldom hold on real wireless networks. Furthermore, we observed that the target loss and delay were often always met with RSF MPTCP where attainable (aggregate achievable loss/delay < target).

## 6. Potential Usage and Applications

The outcomes of this research paper laid the foundation for improving the responsiveness of the future mobile internet. The potential benefits, usage, and applications can be observed from the following three different perspectives.

### 6.1. Benefits to Service Providers, the Environment, and Society

Internet giants, such as Amazon, found that every 100 ms of latency cost them 1% in sales; Google found that an extra 0.5 s in search page generation time dropped traffic by 20%. The gaming industry, where latencies larger than even 80 ms can hurt gameplay, has even tougher latency requirements. The expectations of today's NOW customers continue to grow, and the amount of data generated and accessed is mind-boggling. CISCO has described how 3.2 quintillion bytes of data are generated every day and has stated, that over the last two years alone, 95% of the data in the world were generated. It is clear that the need for speed and scale is escalating, and we need to understand how we can support future networks to remain competitive from all aspects: optimized operations, maximized network utility, and enhanced user experience. The benefits are not purely for large corporations. Reduced latency improves the quality of gaming and video telephony, improving the lives of isolated rural and under-connected global populations. Moreover, the applicability of the outcomes to the internet of vehicles will also help combat traffic congestion, bringing benefits to both the environment and society.

### 6.2. Potential for Impact in 6G Specifications

The 3GPP specification put forward ATSSS for multi-connectivity of WiFi and cellular over the 6G core [2,3]. This is also reflected in the ongoing development by Google of BBR (bottleneck bandwidth and roundtrip time) and the redesign of QUIC (a clean-sheet design overlaid on UDP, standardized by IETF in 2021) at the transport layer. Important trends include: (i) multi-connectivity with ATSSS, especially the concurrent use of both 3GPP and non-3GPP interfaces on mobile devices [34], (ii) use of forward error correction (FEC) to reduce latency, (iii) improving operation over lossy wireless links, and (iv) support for incremental rollout by avoiding the need for changes to the network fabric. However, many knowledge gaps need to be resolved for these to occur and for them to be applied for emerging technologies, which range from semantic communication [35] to federated learning over wireless vehicle-to-grid networks [36].

### 6.3. Applicability of the Proposed RSF Framework

We employed the RSF framework [8] in this paper to guarantee network utility maximization in a contest between many data subflows for future internet applications. The RSF framework can be enhanced and applied to simulate the interplay of rival entities in the field of distributed computing networks [13]. To this end, the competition among social media users for exposure over the timeline was modeled using RSF [12]. In a different context, an RSF-based incentive mechanism for crowd-sourcing was developed [37] and the contention between the miners was framed as RSF [38] in order to examine the multi-cryptocurrency blockchain. Recently, RSF has been employed for strategic resource management to operationalize network slicing [39]. Therefore, the developed RSF framework and the findings in this paper are not only timely, but have the potential for high impact in the field.

## 7. Conclusions

We developed a novel framework for designing MPTCP algorithms by using a *rent-seeking* game-theoretic framework. This approach guarantees the existence of a utility function with strict diagonal concavity. Another advantage of the diagonally strict concave property of the utility function is that it is possible to develop dynamic congestion controls that converge to the normalized equilibrium while guaranteeing stability. Our network emulation experiments have demonstrated higher throughput for our RSF MPTCP implementations without compromising fairness. Furthermore, our RSF MPTCP approach is highly robust to the characteristics of heterogeneous network paths and provides better responsiveness than other existing MPTCP algorithms.

**Author Contributions:** Conceptualization, S.R.P.; Formal analysis, S.R.P.; Investigation, S.R.P.; Methodology, S.R.P.; Resources, S.R.P.; Supervision, C.W. All authors have read and agreed to the published version of the manuscript.

**Funding:** Financial support for this work was provided in part by School of IT Research Grants (Deakin University) and Canada's Natural Sciences and Engineering Research Council (NSERC).

**Data Availability Statement:** Not applicable.

**Acknowledgments:** The authors gratefully thank the anonymous IFIP Performance 2020 reviewers and MDPI Editors for their constructive comments and suggestions on an earlier version of this paper.

**Conflicts of Interest:** The authors declare no conflict of interest.

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
