# Peer review of "A Game-Theoretic Rent-Seeking Framework for Improving Multipath TCP Performance†"

_futureinternet, doi:10.3390/fi14090257_

Round 1

Reviewer 1 Report

The extended version of your work presented at IFIP Performance 2020, November 2-6, 2020, Milan, Italy. Current version of the article uses the same figures and many results of the previous work. If the material already got published by the same authors or different authors, it need to cted not repeating the same thing in the new article. So, it isn't easy to analyse the actual contribution of this particular draft.

The Abstract section has to be accompanied by numerical data or quantitative information, enabling authors to convey their analysis and findings in a more solidified and comprehensive manner. Two or three extra sentences, or the enrichment of the existing narrative with numerical data, they are recommended.

Abstract needs to be improved, it should include the objective of the work, techniques. Also, you need to specify the right applications of the work.

References are poor … No recent or top articles. Citing [11,20,21] and [3,4,6–9,27,29,32,34] is not good practice.

This paper is missing many important thigs and this makes it a poor paper.

  • Is not clear what is the scope of the paper based on the previous work

  • Is it proposing something new? what excatly new incomparision [23]?

  • What is the novelty? Did you complete a deep review of the topic?

  • What is the step ahead made here in comparison with [23]
  • There are too many typos, missing punctuation, missing references

Perhaps a Conclusions section could be useful for emphasizing the outcomes and novel contributions of this research

Author Response

We thank the reviewers very much for their valuable and constructive comments that we believe have helped improve the quality of the paper. In the new version we have settled the major issues, also using the feedback of the other reviewers. In particular, we have provided the reader with more intuition behind the approach followed and demonstrated the theory and dynamics of the network system which is essential for practicable MPTCP protocol. In addition, the experimental setup has been explained clearly. We also included precise definitions of all metrics, and we fixed all typos and errors. Please find below in more detail how we adapted the manuscript. This is full version of our work presented at IFIP Performance 2020, November 2-6, 2020, Milan, Italy. As per the conference policy, the abstract version [23] appears in ACM SIGMETRICS Performance Evaluation Review and the full-length journal version is to be published for clarity on the theory and detailed evaluations.

Reviewer 2 Report

This work is an extension of the author’s work presented in an earlier conference. It contains worth-to-mention results and articulated well. I recommend the publication of this work (with minor modifications).

1.      Don’t start the abstract with “because”. This is not the convention.

2.      Introduction (and the rest) mention about 5G and the Gigabit WiFi network and their throughput and delay-jitter. This is debatable. Being a Layer-2 technology, 5G or Gigabit WiFi does not guarantee bounded jitter, and throughput. They only concern about “one-hop” communication. Higher-layer protocols need to take advantage of certain new features at L2 to have less delay-jitter (again it depends on path congestion) and throughput. Please revise suitably.

3.      Motivation: This is disconnected from the Introductory section, which talk about 5G and WiFi. How do you interconnect with your introductory section?

4.      Paper provides several pages of known information (on stability). Can this be removed, and a suitable reference is made to standard books? (or justify their use here)

5.      Fig 2: What characteristics of the WiFi network is emulated using Ethernet? What are the non-deterministic path-delay characteristics in your simulated model?

Author Response

We thank the reviewer very much for their valuable and constructive comments that we believe have helped improve the quality of the paper. In the new version we have settled the major issues, also using the feedback of the other reviewers.

Reviewer 3 Report

The title as well as the abstract raised expectations about the manuscript and research accomplished. This research would be a relevant addition to existing literature. Thank you for this valuable contribution. The research in the manuscript is an extended version of work presented at an international symposium in 2020.

I will structure my feedback in (a) general remarks (these comments cover feedback applicable in the entire manuscript), and (b) specific remarks (feedback on sentence and/or word level). The specific remarks can include a quote from your original manuscript to refer to a specific section. Some of the specific remarks will refer to specific line number (e.g. L110).

General remarks

This is a well written and well thought manuscript to address the problems with well-defined utility function for existing Multipath TCP algorithms. This paper intends to present theoretical analysis to verify the new RSF MPTCP method proposed. For researchers, this manuscript provides network emulation testbeds and various scenarios considering main network output parameters, the results were promising with greater throughput, better resilience, and increased responsiveness compared to competing MPTCP solutions. The paper is well supported by mathematical equations and pseudo codes, but the paper requires checking for English and formatting.

Specific remarks

Clearly highlight the novelty in the abstract. For example, L 174- the sentence “we propose a new MPTCP algorithm that generalizes existing algorithms, and strikes a good balance among these properties” is a good one to include in the abstract

L155-rephrase the sentence, maybe move however to the beginning.

L 203-are 3 and 4 references? If yes, why they were put in bracket [18, (3)-(4)]

L236- elaborate what Thm. 4 means in [27, Thm. 4]

L257 and L301- typo mistakes; squares appear in the text,

For Table 1, it is much neater if a new field with the title “MPTCP algorithms” is added

Figure 1 is far from the related text

Third person is better for Figure 2 title, it is better to take away “we”.

L428-it is better to change “Observation 2/3” to “Observation” subsection for each scenario  

References should be numbered in the orders they appeared in the text.

L553- section title is missing “References”

follow the journal template and check the reference list for inconsistencies in spaces, commas, and year boldface, and italic font.

Author Response

We thank the reviewer very much for their valuable and constructive comments that we believe have helped improve the quality of the paper. I

Round 2

Reviewer 1 Report

The authors should stress out also the usage and the applications of the paper main findings indicating the novelty brought by the presented model.

Author Response

Response: We thank the reviewer for this valuable and constructive suggestion. The outcomes of this research paper laid the foundation to improve the responsiveness of the future mobile internet. The usage and applications of the main findings in this work in connection with the novelties have been discussed from three different perspectives in the revised version.

i) Benefits to service providers, environment and society

ii) Potential for impact in 6G Specification

iii) Applicability of the proposed RSF Framework

Modification: We have added a new section (Sec 6, please see the attached version) and discussed the aforementioned three aspects in detail in the revised version of the manuscript.
